# Antimicrobial Activity and Mode of Action of N-Heterocyclic Carbene Silver(I) Complexes

**DOI:** 10.3390/molecules30010076

**Published:** 2024-12-28

**Authors:** Giusy Castagliuolo, Michela Di Napoli, Tshering Zangmo, Joanna Szpunar, Luisa Ronga, Anna Zanfardino, Mario Varcamonti, Diego Tesauro

**Affiliations:** 1Department of Biology, University of Naples “Federico II”, Via Cynthia 26, 80126 Naples, Italy; giusy.castagliuolo@unina.it (G.C.); michela.dinapoli@unina.it (M.D.N.); varcamon@unina.it (M.V.); 2IPREM (Institut des Sciences Analytiques et de Physico-Chimie pour l’Environnement et les Matériaux), CNRS, E2S UPPA, Université de Pau et des Pays de l’Adour, CEDEX 9, 64012 Pau, France; tshering.zangmo@univ-pau.fr (T.Z.); joanna.szpunar@univ-pau.fr (J.S.); luisa.ronga@univ-pau.fr (L.R.); 3Department of Pharmacy and Interuniversity Research Centre on Bioactive Peptides (CIRPeB), University of Naples “Federico II”, Via Montesano 49, 80131 Naples, Italy; dtesauro@unina.it

**Keywords:** NHC silver complexes, Gram (+) and Gram (−) bacteria, antimicrobial properties, SEC-ICP-MS

## Abstract

Silver drugs have played a vital role in human healthcare for the treatment of infections for many centuries. Currently, due to antibiotic resistance, a potential scenario or the application of silver complexes may arise as substitutes for conventional antibiotics. In this perspective, N-heterocyclic carbene (NHC) ligands have been selected as carrier molecules for silver ions. In this study, we selected two mono NHC-silver halide complexes: bromo[1,3-diethyl-4,5-bis(4-methoxyphenyl)imidazol-2-ylidene]silver(I) (Ag4MC) and chloro[2-pyridin- N-(2-ethylacetylamido)-2-yl-2*H*-imidazol-2-ylidene]silver(I) (Ag5MC), and two cationic bis NHC silver complexes: bis[1,3-diethyl-4,5-bis(4-methoxyphenyl)imidazol-2-ylidene]silver(I) (Ag4BC) and bis[2-pyridin-N-(2-ethylacetylamido)-2-yl-2*H*-imidazol-2-ylidene]silver(I) (Ag5BC). The inhibitory properties of the four complexes were evaluated for their antimicrobial potential against a set of Gram (+) and Gram (−) bacterial strains and the fungus *C. albicans*. In addition, further investigations were made using fluorescence and scanning electron microscopy (SEM) in order to gain more insights into the mechanism of action. Some preliminary information on the Ag target was obtained by analyzing the cytosol of *E. coli* treated with Ag5MC by size-exclusion chromatography (SEC) coupled with inductively coupled plasma mass spectrometry (ICP-MS).

## 1. Introduction

Antimicrobial resistance (AMR) is a worrying global health menace [1]. AMR is hard to fight, with high potential costs in terms of human lives and from an economic point of view [2]. One promising approach is the use of transition metals because they exhibit fast and noteworthy activity, at low concentrations, in prokaryotic cells.

Since ancient times, silver has been known to be effective against a broad range of microorganisms, but knowledge of its mechanism of action is limited. The silver cation acts by binding biomolecules due its Lewis acid properties [3]. Silver cations can be generated by the release of silver nanoparticles or from silver complexes by loss of ligands. The ancillary ligands are able to tune the reactivity and address Ag(I) toward the target. Since the development of the first *N*-Heterocyclic Carbenes (NHCs) ligand in 1991 [4], this class of ligand has been employed to coordinate silver cations [5]. The widespread use and applications of this moiety can be attributed to the simplicity of its synthetic methods and stability of the carbene center, which is stabilized by the presence of two adjacent nitrogen atoms [6,7].

The antimicrobial and anticancer properties of this class of complexes have been reviewed in the last decade [8,9,10]. It is critical in the design to achieve complexes that release the ions into the affected area while maintaining efficacy at the wound site.

In actuality, the NHC ligand structures considerably affect activities of the NHC–silver complexes. Factors such as hydrophobic substitution and steric bulk on the imidazole ring can delay the rate of silver ions release [11]. The presence of lipophilic benzyl-substituents at the *N*1 and *N*3 positions on 4,5-diphenylimidazole allowed the recording of a low minimum inhibitory concentration (MIC), ranging from 20 to 3.13 µg/mL (35.3 to 5.52 µM), for a variety of Gram-positive, Gram-negative, and mycobacteria tested [12]. The choice of NHC ligand in silver complexes is also critical to the mode of action against bacteria. If silver ions are released in extracellular media, they can damage bacterial membranes, causing cell protein denaturation. The stability of the complex allows it to cross the membrane and the silver ion can interact with the amino acid residues of proteins inside the cytosol. Thiol groups, carboxylates, phosphates, hydroxyls, imidazoles, indoles, and amines, either singly or in combination, can be linked so that multiple poisonous events, rather than specific lesions, simultaneously interfere with microbial processes. Molecular targets remain largely unknown, despite the long-time use of silver as an antimicrobial agent. In recent years, different attempts have been made to find the predominant target. Many studies have been conducted to investigate the action mechanism of silver nanoparticles [13,14], while silver complexes have not been explored in depth.

Some hypotheses and insights have recently been proposed. Sun et al. delineated the first dynamic antimicrobial actions of silver ion in *E. coli* by using integrated omic approaches, including metalloproteomics, metabolomics, bioinformatics, and systemic biology [15]. These studies have allowed us to establish that the Ag ion primarily damages multiple enzymes in glycolysis and the tricarboxylic acid (TCA) cycle, leading to the stalling of the oxidative branch of the TCA cycle and an adaptive metabolic divergence to the reductive glyoxylate pathway.

The same group identified six silver-binding proteins in *E. coli*, coupling gel electrophoresis with inductively coupled plasma mass spectrometry (GE-ICP-MS) [16]. Among them, they demonstrated that silver inhibits the enzymatic function of glyceraldehyde-3-phosphate dehydrogenase (GAPDH) through targeting Cys149 in the catalytic site. Another site of interaction can be the *E. coli* malate dehydrogenase (EcMDH) enzyme. This enzyme includes three cysteine-containing sites, as demonstrated by X-ray crystallography [17]. More recently, 38 Ag-binding proteins were separated and identified in *Staphylococcus aureus* at the whole-cell scale. Among these, it was validated that silver ion could inhibit a key target 6-phosphogluconate dehydrogenase through binding to catalytic His185 [18]. All these studies were carried out by administering silver salts. Very recently, Sahin et al. analyzed, by molecular docking methods, the potential target of a series of NHC–Ag complexes, such as N-acyl homoserine lactone (AHL) lactonase [19].

Therefore, it is necessary to obtain more information about the appropriate characteristics of the NHC ligand and search for possible Ag-targets. The aim of this study is to test new NHC–Ag complexes, bearing lipophilic and functional groups, and reach more insights about their action.

For this purpose, we selected and synthesized two halide mono NHC-silver complexes: bromo[1,3-diethyl-4,5-bis(4-methoxyphenyl)imidazol-2-ylidene]silver(I) (Ag4MC) and chloro[2-pyridin- N-(2-ethylacetylamido)-2-yl-2*H*-imidazol-2-ylidene]silver(I) (Ag5MC), and two bis NHC silver cationic complexes: bis[1,3-diethyl-4,5-bis(4-methoxyphenyl)imidazol-2-ylidene]silver(I) (Ag4BC) and bis[2-pyridin- N-(2-ethylacetylamido)-2-yl-2*H*-imidazol-2-ylidene]silver(I) (Ag5BC) (Figure 1). We studied the inhibitor effect of the four silver complexes on Gram (−) and Gram (+) bacterial strains and the fungus *Candida albicans*. Therefore, we performed fluorescence microscopy (FM) and scanning electron microscopy (SEM) studies on *E. coli* and *S. aureus* to explore potential bacterial targets.

Some preliminary insights into the potential Ag-cellular target were achieved by analyzing the cytosol of *E. coli* treated with Ag5MC by size-exclusion chromatography (SEC) coupled with inductively coupled plasma mass spectrometry (ICP-MS).

## 2. Results and Discussion

### 2.1. Selection and Synthesis of the Complexes

The selection of the complexes was based on the search for lipophilic ligands and the presence of specific functional groups, e.g., the amide. The crucial point was to guarantee a slow release of the silver cation, improving the stability of Ag(I)–NHC complexes. Substituents at the 4 and 5 positions of the imidazolin-2-ylidene ligand can ensure this task. For these statements, we selected the bromo[1,3-diethyl-4,5-bis(4-methoxyphenyl)imidazol-2-ylidene]silver(I) (Ag4MC) complex [20]. Liu et al. demonstrated that this complex has high cytotoxicity on different cancer cell lines, due to the high accumulation induced by lipophilic properties of the 4,5 aryl substituents on the imidazole ring of the ligand [20]. They did not report the eventual antibacterial properties of this compound. The cationic (bis[1,3-diethyl-4,5-bis(4-methoxyphenyl)imidazol-2-ylidene]silver(I)(Ag4BC) complex was prepared to compare the activity of the positive charge of this compound to Ag4MC [20]. Both complexes Ag4MC and Ag4BC were synthesized as reported in the literature [20].

Chloro[2-pyridin- N-(2-ethylacetylamido)-2-yl-2*H*-imidazol-2-ylidene]silver(I) (Ag5MC) and chloro[2-pyridin- N-(2-ethylacetylamido)-2-yl-2*H*-imidazol-2-ylidene]silver (I) (Ag5BC) were designed to retain the lipophilic behavior and introduce an amide function.

The ligand 2-(2-acetamidoethyl)-2*H*-imidazo[1,5-*a*]pyridin-4-ium as Cl^−^ salt (5MC) was synthesized by adapting a previously reported procedure [21]. The synthesis of 2-(2-acetamidoethyl)-2*H*-imidazo[1,5-a]pyridin-4-ylium chloride was optimized (Figure 1). In the first step, the primary amine reacted with paraformaldehyde, and then 2-pyridinecarboxaldehyde was added, affording the 2-(2-acetamidoethyl)-2*H*-imidazo[1,5-*a*]pyridin-4-ium as Cl^−^ salt. The synthesis of the Ag5MC silver complex was performed, adding to the salt Ag_2_O in the CH_3_OH/CH_2_Cl_2_ mixture. The formation of the Ag5BC was achieved similarly after exchanging Cl^−^ for a non-coordinating PF_6_^−^ anion.

### 2.2. Antimicrobial Activity

Cell survival assays were performed to investigate the antimicrobial properties of the silver compounds Ag4BC, Ag4MC, Ag5BC, and Ag5MC. The selected indicator organisms were Gram (−) *E. coli*, Gram (+) *S. aureus*, and the fungus *C. albicans*. All species were incubated with a fixed concentration of 5 µM for each complex in order to assess survival percent trends. Ligands of the individual compounds were tested as controls at the same concentration.

The Ag5MC complex exhibited the highest antimicrobial activity against the tested strains, as shown in Figure 2. A significant reduction in bacterial survival (expressed as a percentage on the *y*-axis) was observed for all three strains, with *E. coli* showing the most pronounced effect. At 5 µM, *E. coli* survival dropped to 0%, while *S. aureus* and *C. albicans* displayed some resistance, with survival rates between 40% and 50%.

As shown in panel A, the ligands did not exhibit any intrinsic antimicrobial properties.

As Ag5MC demonstrated excellent antimicrobial potential, comparable to Ag_2_O, used as a positive control, we focused our study on analyzing the antimicrobial properties of this compound. We tested a range of concentrations of the 5MC ligand and the Ag5MC complex (0, 0.25, 0.5, 1, 2.5, 5, 10, and 20 µM) against the same strains using the same previous bacterial survival assay.

As shown in Figure 3, *the* data confirm the previous findings: *E. coli* is the most sensitive strain to Ag5MC, exhibiting a much more pronounced dose-dependent response compared to *S. aureus* and *C. albicans*. Notably, the black bar representing *E. coli* survival disappears completely after treatment with 5 µM, while the two gray bars (representing *S. aureus* and *C. albicans*) only disappear at 20 µM. These results highlight that Ag5MC is more effective against Gram (−) bacteria than against fungi and Gram (+) bacteria.

The data obtained after treatment with the ligand further confirmed the previous findings: even at a concentration of 20 µM, the free ligand did not exhibit any antimicrobial activity, with bacterial survival remaining at 100%.

To further investigate this observation, we decided to determine the minimum inhibitory concentration (MIC) of Ag4BC, Ag4MC, Ag5BC, and Ag5MC, extending the study to include additional Gram (+) and Gram (−) bacterial strains.

The MIC was determined by using the microdilution method; the results are reported in Table 1. Consistent with previous experiments, the lowest MIC values (80–108 µM) were observed for Ag5MC against Gram (−) strains, as well as some Gram (+) strains (100–200 µM). This confirms that Ag5MC exhibits the strongest antimicrobial activity among all the silver compounds tested, with a more pronounced effect against Gram (−) bacteria. *E. coli* consistently proved to be the most sensitive strain.

To gain a more comprehensive understanding of the bacterial toxicity of the selected compounds, we evaluated bacterial cell viability using the MTT assay, testing all ligands and compounds against model strains: *E. coli*, *S. aureus*, and *C. albicans*.

As shown in Figure 4, the percentage of bacterial viability is comparable to the positive control (untreated cells) for all samples, except for Ag5MC. This compound was confirmed to be toxic to all selected strains, showing 0% viability after treatment at a concentration of 100 µM.

### 2.3. Characterization of the Compound Ag5MC Mechanism of Action

Since the cellular target of silver compounds is not well established in previous studies, we decided to investigate the possible mechanism of action of Ag5MC responsible for its antimicrobial activity using microscopy techniques.

To explore potential bacterial targets, we performed fluorescence microscopy (FM) studies on *E. coli* and *S. aureus*, using 4′,6-diamidino-2-phenylindole dihydrochloride (DAPI) and propidium iodide (PI). DAPI freely crosses the cell membrane, binds to DNA, and stains cells blue, while PI cannot penetrate intact bacterial membranes, staining only cells with damaged membranes red.

After 4 h of treatment with Ag5MC (60 µM), fluorescence microscopy revealed that the complex did not appear to cause membrane damage (Figure 5). Strong blue fluorescence was observed for cells treated with Ag5MC (B-2; D-4), as well as control cells (A-1; C-3), indicating intact membranes. This analysis suggests that the antimicrobial action of Ag5MC against *E. coli* and *S. aureus* may not involve membrane disruption, but is likely to target other cellular mechanisms, leading to a reduction in bacterial cell survival.

To further investigate using microscopy analysis, we performed an additional study using scanning electron microscopy (SEM). To examine the antimicrobial activity of Ag5MC via SEM, we replicated the experimental conditions from the previous study—treating *E. coli* with 60 µM of Ag5MC for 4 h.

SEM observations confirmed that the cellular membrane was not the primary target of Ag5MC (Figure 6). Both untreated cells (A) and Ag5MC-treated cells (B) appeared intact. However, the treated samples clearly showed significant reduction in the number of bacterial cells, further supporting the hypothesis that Ag5MC affects other cellular mechanisms rather than directly damaging the membrane.

To confirm that the mechanism of action of Ag5MC does not involve membrane disruption, but likely affects other cellular processes, we performed an additional test using the NPN probe. 1-N-phenyl naphthylamine is a hydrophobic fluorescent probe with weak fluorescence in an aqueous environment and strong fluorescence in hydrophobic environment, such as within the membrane. Normally, it cannot enter cells, as it does not penetrate the outer membrane. However, when the outer membrane is damaged, NPN can enter and emits a strong fluorescence, indicating membrane disruption.

*E. coli* and *S. aureus* cells were incubated for 30 min with the NPN probe (40 μM) and Ag5MC (60, 100, and 200 µM), with SDS (0.1%) used as a positive control. As shown in Figure 7, SDS causes an increase in fluorescence intensity, indicating bacterial membrane damage. In contrast, both *E. coli* (panel A) and *S. aureus* (panel B) cells showed no significant increase in NPN fluorescence intensity, even after treatment with Ag5MC at 200 µM, confirming that the target of this compound is not the bacterial membrane.

### 2.4. MTT on HaCaT Cells Assay

Cell survival experiments were conducted on HaCaT cells (immortalized human keratinocytes) at various concentrations of Ag5MC (0, 1, 10, 50, and 100 µM). According to the MTT assay, after 4, 24, and 48 h of incubation with Ag5MC, the treated cells showed viability comparable to the control group up to a concentration of 50 µM. These results indicate that Ag5MC starts to be cytotoxic when it reaches a concentration of 100 µM and based on exposure time (Figure 8). The LD_50_ concentration of Ag5MC demonstrates remarkable selectivity, being approximately 20-fold lower when tested against *S. aureus* and *C. albicans* compared to its effect on human HaCaT cells. Furthermore, the compound shows even greater specificity against *E. coli*, with an LD_50_ concentration that is over 200-fold lower than that observed for HaCaT cells. These results suggest that Ag5MC possesses significant antimicrobial potency, while maintaining a relatively lower cytotoxicity profile towards human cells, underlining its potential suitability for therapeutic applications.

### 2.5. SEC-ICP-MS Analysis

Treated and untreated *E. coli* cultures with Ag5MC were subjected to sonication to obtain cytosolic proteins from the cell lysates. The extracted proteins were run on a 12% SDS-PAGE gel and subsequently quantified using the BCA assay. The cytosolic protein fraction treated with Ag5MC was analyzed by size-exclusion chromatography (SEC) coupled with inductively coupled plasma mass spectrometry (ICP-MS). The completeness of the species elution (column recovery) was controlled and proved to be quantitative. Figure 9 shows the SEC-ICP chromatogram of cytosol treated with Ag5MC producing mainly three poorly resolved Ag containing peaks, suggesting the interaction of proteins with Ag5MC. The molecular weight corresponding to the most abundant species is higher than ca. 100 kDa, with the MW of the other ones being smaller, but still higher than 50 kDa.

In order to further investigate the presence of Ag binding species, the untreated cytosol was exposed to an excess of Ag^+^ (based on the concentration of Ag found to be 110 µg/L ± 4.2 µg/L in the treated cytosol) and then analyzed by SEC-ICP-MS. The relative chromatogram is superposed to that of NHC–Ag(I)-treated cytosol in Figure 10. Two main peaks appeared after the incubation of cytosol with Ag^+^ at the same elution volume as for the treated cytosol (at ca. 9 and 10.2 min), indicating that proteins in the same mass ranges interacted with silver ion. However, an additional peak was detected in the treated cytosol chromatogram (at ca. 9.7 min, highlighted by an arrow); it was absent in the incubated one. This suggests that the interaction of the cytosol biomolecules with Ag5MC differs from that with Ag^+^ ion, or that in vivo interaction involves a higher number of proteins.

## 3. Experimental

Solvent and chemical reagents were purchased and used as received without further purification (Sigma-Aldrich, Steinheim, Germany); ^1^H NMR spectra were acquired with Brucker 400 MHz and Brucker 700 MHz; ^13^C NMR were acquired with Brucker 100 MHz and Brucker 175 MHz. The FT-IR spectra were obtained by using a Ge single crystal at a resolution of 4 cm^−1^ in attenuated total reflection mode on a Jasco FT/IR 4100 spectrometer (Easton, MD, USA). The identity of complexes was assessed by mass spectrometry using an LTQ XL Linear, ion trap mass spectrometer, ESI source (Finnigan/Thermo Electron Corporation, San Jose, CA, USA). The complexes bromo[1,3-diethyl-4,5-bis(4-methoxyphenyl)imidazol-2-ylidene]silver(I) (Ag4MC) and bis[1,3-diethyl-4,5-bis(4-methoxyphenyl)imidazol-2-ylidene]silver(I) (Ag4BC) were synthesized as reported in the literature [20]. ESI mass and ^1^HNMR spectra are reported in the Appendix A.

### 3.1. Syntheses

#### 3.1.1. 2-(2-Acetamidoethyl)-2H-imidazo[1,5-a]pyridin-4-ylium Chloride (5MC)

Paraformaldehyde (45.0 mg, 1.50 mmol) was added to a stirring solution of N-acethylendiamine (102 mg, 1.00 mmol) in ethanol (2.00 mL). The resulting mixture was stirred at room temperature for 4 h, at which point the solution became homogeneous. Afterwards, 3.00 mol L^−1^ HCl in CH_3_CH_2_OH (0.333 mL) and 2-pyridinecarboxaldehyde (107 mg, 1 mmol) were added and the resulting mixture was stirred at room temperature overnight. Then, the solvent was removed under reduced pressure. Purification of the residue on silica gel (CH_2_Cl_2_/CH_3_OH 8.5/1.5) afforded the product as a colorless oil (110 mg, yield: 48.0%).

^1^H NMR (400 MHz, CD_3_OD-d_4_) δ 8.48 (d, J = 7.2 Hz, 1H), 8.16 (s, 1H), 7.88 (d, J = 10.4 Hz, 1H), 7.27 (ddd, J = 9.3, 6.6, 1.0 Hz, 1H), 7.19 (td, J = 6.9, 1.2 Hz, 1H), 4.56–4.51 (m, 2H), 3.75 (q, J = 5.8 Hz, 2H), 1.79 (s, 3H). ^13^C NMR (100 MHz, CD_3_OD-d_4_) δ 172.4 (C=O), 130.2, 125.0, 123.5, 118.2, 118.0, 113.0, 50.0, 39.2 (NH-CH_2_) 20.0 (CH_3_). (ESI+) = 204 [M- Cl+H]^+^. C, 55.41; H, 5.92; N, 17.63; Found C, 55.42; H, 5.95; N, 17.61.

#### 3.1.2. Synthesis of 2-(2-Acetamidoethyl)-2H-imidazo[1,5-a]pyridin-4-ylium Silver(I) Chloride (Ag5MC)

Added to a stirred solution of 2-pyridin- N-(2-ethylacetylamido)-2-yl-2-imidazole chloride (60 mg, 0.25 mmol) in 11.0 mL of dichloromethane/methanol (6:1) was silver(I) oxide (37 mg, 0.101 mmol) under N_2_, and the reaction mixture was allowed to stir for 6 h in the darkness. The precipitate was removed filtering over a bed of Celite. The filtrate was concentrated under vacuum and the residue was precipitated in dichloromethane/ethyl ether achieving a white solid (50 mg, yield: 58%). ^1^H NMR (700 MHz, CD_3_OD-d_4_) δ 8.60 (d, J = 7.2 Hz, 1H), 7.80 (s, 1H), 7.55 (d, J = 10.4 Hz, 1H), 7.05 (ddd, J = 9.3, 6.6, 1.0 Hz, 1H), 6.80 (td, J = 6.9, 1.2 Hz, 1H), 4.70 (m, 2H), 3.75 (q, J = 5.8 Hz, 2H), 1.79 (s, 3H). ^13^C NMR (175 MHz, CD_3_OD-d_4_) δ 173.0 (C=O), 132.0, 128.5, 123.8, 118.0, 114.5, 111.8, 52.0 (N-CH_2_), 39.2 (NH-CH_2_) 21.0 (CH_3_). (ESI+) = 346 [M+H]^+^. C 38.18; H, 4.08; N,12.15; Found C, 38.20; H, 4.09; N, 12.10 ESI+-MS, m/z: 346 [M+H]^+^).

#### 3.1.3. Synthesis of 2-Pyridin- N-(2-ethylacetylamido)-2-yl-2-imidazole Hexafluorophosphate (5BC)

To a solution of 2-pyridin- N-(2-ethylacetylamido)-2-yl-2-imidazole, chloride (0.187 mmol) in 1.5 mL of water and 0.210 mmol of KPF_6_ were added dropwise and stirred for 14 h at room temperature. The solution was extracted with 3 portions of ethyl acetate and the combined organic layer was washed with brine, dried over sodium sulfate, and dried under vacuum. The residue was washed with little portions of diethyl ether, achieving the product color of pale yellow solid (49 mg, yield: 67%) ^31^P NMR -148 (septet J = 711 Hz) ^19^F NMR -74.0 (d, J = 711 Hz) ESI+-MS, m/z: 204 [M-PF_6_+H]^+^.

#### 3.1.4. Synthesis of 2-Pyridin- N-(2-ethylacetylamido)-2-yl-2-imidazole Silver(I)Hexafluorophosphate (Ag5BC)

To a stirred suspension of 2-pyridin- N-(2-ethylacetylamido)-2-yl-2-imidazole Hexafluorophosphate (50 mg, 0.14 mmol) in 20 mL of dichloromethane/methanol (1:1) were added silver(I) oxide (32 mg, 0.071 mmol), tetrabutylammonium bromide (TBAB) (46 mg, 0.14 mmol), and 0.700 mL of a solution of NaOH 1 mol L^−1^. The reaction mixture was allowed to stir for 4 h in the darkness at room temperature. The precipitate was removed by filtering over a bed of Celite. The filtrate was evaporated to dryness. The residue was washed with ethyl ether and the product recovered as pale yellow solid (22 mg, yield: 48%).

^1^H NMR (400 MHz, CD_3_OD-d_4_) δ 8.55 (m, 2H), 7.75–7.60 (m, 2H), 7.53 (s, 2H), 7.05–7.00 (m, 2H), 6.80–6.70 (m, 2H), 4.65 (m, 4H), 3.75 (m, 4H), 1.85 (s, 6H). ^13^C NMR (175 MHz, CD_3_OD-d_4_) δ 173.0 (C=O), 132.0, 128.5, 123.8, 118.0, 114.8, 111.7, 52.2, 39.7 (NH-CH_2_) 21.5 (CH_3_). C,40.10; H,4.28; N,12.75; Found C, 40.12; H,4.27; N,12.77 (ESI+) = 514 [M+H]^+^.

### 3.2. Bacterial Strains

The complexes were evaluated against the Gram (−) bacterial strains (*E. coli* DH5α, *Pseudomonas aeruginosa* PAO1 ATCC 15692, *Shigella sonnei* ATCC25931 and *Salmonella* Typhimurium ATCC14028) and the Gram (+) bacterial strains (*S. aureus* ATCC6538P, *Bacillus subtilis* AZ54, *Streptococcus mutans* ATCC 35668 and *Streptococcus oralis* CECT 8313). The complexes were also tested against the fungus *C. albicans* ATCC 14053.

### 3.3. Antimicrobial Assay

The antimicrobial activity was assessed by measuring the cell viability of Gram (−), *E. coli*, Gram (+), *S. aureus*, model strains, and the fungus *C. albicans*. Bacterial cells were initially exposed to a 5 µM concentration of each ligand (4BC, 4MC, 5BC, and 5MC) as well as the corresponding compounds Ag4BC, Ag4MC, Ag5BC, and Ag5MC. Untreated cells served as the negative control, and Ag_2_O-treated cells served as the positive control. After 4 h of incubation at 37 °C with shaking at 150 rpm, serial dilutions (1:10, 1:100, 1:1000) of all samples were prepared and plated on LB agar in Petri dishes, which were incubated overnight at 37 °C. Bacterial survival was determined the following day by counting the colonies [22].

Subsequently, following the same procedure, the antimicrobial activity of the 5MC and Ag5MC component was investigated in detail, with only the concentrations being changed to 0, 0.25, 0.5, 1, 2.5, 5, 10, and 20 µM.

### 3.4. Determination of Minimal Inhibitory Concentration

To determinate the minimal inhibitory concentrations (MICs) of Ag4BC, Ag4MC, Ag5BC, and Ag5MC against all bacteria, we used the microdilution method established by the Clinical and Laboratory Standards Institute (CLSI), according to Di Napoli et al. [23] A total of ~5 × 10^5^ CFU/mL were added to 95 µL Mueller–Hinton broth (CAM-HB; Difco), with or without the different samples at different concentrations (0.1–200 µM). After overnight incubation at 37 °C, MIC_100_ values were determined as the lowest concentration responsible for no visible bacterial growth, reading the OD at 600 nm. The positive control was represented by Ag_2_O.

### 3.5. Fluorescence Microscopy Analyses: DAPI/PI

*E. coli* and *S. aureus* cells were incubated in the darkness with shaking for 4 h at 37 °C, in the presence or absence of Ag5MC at a sub-MIC concentration of 60 µM. Following incubation, 10 µL of sample was treated with 4′,6-diamidino-2-phenylindole dihydrochloride (DAPI) at a concentration of 1 μg/mL and propidium iodide (PI) at a concentration of 10 μg/mL. The samples were observed using an Olympus BX51 fluorescence microscope (Olympus, Tokyo, Japan) with a DAPI filter (excitation/emission: 358/461 nm). Standard acquisition times for DAPI/PI dual staining were set at 1000 ms. Images were captured using an Olympus DP70 digital camera [24].

### 3.6. 1-N-Phenyl Naphthylamine (NPN) Assay

The outer membrane (OM) permeabilizing activity of Ag5MC was assessed using the NPN (1-N-phenylnaphthylamine) assay, with slight modifications based on the method by Jia et al. [25]. Briefly, *E. coli* and *S. aureus* cells were harvested after overnight growth, washed, and resuspended in 5 mM N-2-hydroxyethylpiperazine-N’-ethanesulfonic acid (HEPES), pH 7.2, to an optical density at 600 nm (OD600) of 0.5 ± 0.02. Then, 50 µL of Ag5MC (at concentrations of 60, 100, and 200 µM) was added to 100 µL of the cell suspension, along with 50 µL of NPN (40 µM concentration), in black 96-well fluorometric plates. The control group consisted of HEPES buffer, bacterial cells, and NPN only. The excitation and emission wavelengths were set to 350 nm and 420 nm, respectively.

### 3.7. Scanning Electron Microscopy Analyses

To examine the effects of Ag5MC on *E. coli* cells, scanning electron microscopy (SEM) was employed. Bacterial cells were incubated for 4 h at 37 °C in the presence or absence of Ag5MC at a sub-MIC concentration of 60 µM. After incubation, the cultures were centrifuged at 7000× *g* for 15 min at 4 °C, and then fixed with 3% glutaraldehyde in phosphate buffer (pH 7.2–7.4) for 2 h at room temperature. The samples were post-fixed with 1% osmium tetroxide in the same phosphate buffer for 1.5 h, followed by complete dehydration with ethanol and critical point drying. The samples were then mounted on aluminum stubs, coated with a thin layer of gold using an Edward E306 evaporator, and observed under an FEI Quanta 200 ESEM (Hillsboro, OR, USA) in high-vacuum mode (P 70 Pa, HV 30 kV, WD 10 mm, spot 3.0) [26].

### 3.8. Cell Viability Assay

The cytotoxicity of Ag5MC in bacterial cells (*E. coli*, *S. aureus*, and *C. albicans*) and in the HaCaT eukaryotic cell line was evaluated using an MTT-based colorimetric assay. In this experiment, 5 × 10^5^ CFU/mL of bacterial cells and 20,000 eukaryotic cells were seeded in 96-well microplates and incubated for 24 h. Bacterial cells were incubated at 37 °C in ambient oxygen, while eukaryotic cells were maintained in a humidified atmosphere of 5% CO_2_ at 37 °C. After incubation, various concentrations of Ag5MC (1, 10, 50, and 100 µM) were added to each well. The HaCaT cell plates were then incubated for 4, 24, and 48 h, and the bacterial plates for 24 h under the same conditions.

To assess cell viability, 10 µL of MTT solution was added to 90 µL of DMEM for eukaryotic cells and 90 µL of Mueller–Hinton broth (CAM-HB; Difco) for prokaryotic cells, followed by a 4 h incubation at 37 °C. After removing the medium, 100 µL of DMSO was added to each well, and plates were incubated for an additional 10 min. Absorbance was measured at 570 nm using a Synergy H4 hybrid microplate reader (Agilent, Santa Clara, CA, USA). Each concentration was tested in quadruplicate, and the experiment was repeated three times. Cell viability was calculated relative to untreated cells (DMSO control), which were considered to represent 100% viability [22,27].

### 3.9. Bacterial Protein Extraction

A single colony of *E. coli* was cultured in LB broth overnight at 37 °C with shaking. The cells were then diluted 1:100 in LB medium and cultured for approximately 2–3 h until an OD600 of 0.3 was reached. Untreated cultures and cultures treated with Ag5MC at a concentration of 60 μM were incubated for 4 h at the same temperature. After incubation, the cells were harvested by centrifugation (4500× *g*, 15 min at 4 °C) and washed three times with cold phosphate-buffered saline (PBS).

The collected pellets were resuspended in PBS buffer and lysed by sonication (amplitude: 20%, 5 s on, 20 s off, for a total of 5 min in an ice-cold water bath). The mixed suspension after sonication was centrifuged to obtain the supernatant. The supernatant was further centrifuged (10 min, 100,000× *g*, 4 °C), fractionated, and subjected to SDS-PAGE, followed by quantification using the Pierce™ BCA Protein Assay Kit (Thermo Fisher Scientific, Waltham, Massachusetts, US) [28].

### 3.10. SEC-ICP-MS Analysis

Semi-quantitative evaluation of the total silver content was carried out by flow-injection analysis using external calibration. A stock-standard solution containing 1000 µg/mL Ag was used in preparing calibration standards. The calibration solutions were prepared using 30 mM Tris-HCl, pH 7.5. The calibration curve was made from 5 points and the blank.

The size-exclusion chromatograms were obtained using an HPLC 1260 Infinity (Agilent, Santa Clara, CA, USA) system coupled with an ICPMS 7700 Series (Tokyo, Japan). The column used was a pre-packed Superdex™ 75 Increase 10/300 GL with particle size 13 µm and a separation range (M_r_) of 3000–70,000 Da (GE Healthcare, Uppsala, Sweden). Protein standards used to calibrate the column were thyroglobulin (660 kDa), ferritin (480 kDa), bovine serum albumin (66.5 kDa), fetuin (48 kDa) ovalbumin (45 kDa), carbonic anhydrase (30 kDa), and hydroxocobalamin (1 kDa), at 1 mg/mL each (Sigma-Aldrich). The isocratic elution with 30 mM Tris-HCl buffer (pH 7.5) mobile phase at a flow rate of 0.7 mL/min was used. The sample injection volume was 50 µL

The operating parameters of the ICP-MS were set as follows: radio frequency power of 1600 W, auxiliary gas flow rate of 1.2 L/min, and nebulizer gas flow rate of 1.0 L/min. ^107^Ag and ^109^Ag isotopes were monitored.

Before the analysis, the cytosols were diluted with 30 mM Tris-HCl, pH 7.5 (1/1, *v*/*v*). The untreated cytosol was incubated for 18 h at 37 °C with silver standard solution (1000 µg/mL) in 4% HNO_3_ (SCP SCIENCE) at an Ag^+^ final concentration of 0.25 µg/mL and then diluted with 30 mM Tris-HCl, pH 7.5 (1/1, *v*/*v*), before analysis. 

## 4. Conclusions

Four NHC–Ag complexes were synthesized and tested against Gram-negative and Gram-positive bacteria strains. The inactivity of Ag4MC and Ag4BC could be attributed to the high stability of the NHC–Ag sigma bond. The equilibrium between the stability of complexes and the ability to release the Ag ion is crucial in NHC–Ag complexes. Among all the tested silver complexes, only the compound Ag5MC showed inhibition of antimicrobial activity, with a more marked effect against Gram (−) bacteria. Ag5BC showed limited activity against *E coli* and generally did not inhibit other bacterial strains. The effectiveness of Ag5MC, greater than that of Ag5BC, is in line with the results obtained for other NHC–Ag complexes [29]. The lack of the positive charge could hinder its internalization in prokaryotic cells. The membrane crossing is decisive for the activity of Ag5MC. The same compound is not able to cross the eukaryotic membrane; therefore, it was not toxic for this cell line under the experimental conditions. Ag5MC crosses the *E coli* membrane without causing damage, as demonstrated by FM and SEM microscopy and from the NPN probe test. Therefore, the activity of the complex is carried out by interacting with the biomolecules present in the cytosol. The cell lysate was analyzed to confirm the silver ion interaction with the protein in cytosol. The SEC-ICP-MS demonstrated the presence of silver bound to biomolecules with MW higher than ca. 100 kDa and other smaller biomolecules, but still higher than 50 kDa. The same lysate treated with silver nitrate showed a different target. These results emphasize the role of the ligands coordinated to the complexes in target selection.

Further experiments will be designed to identify protein targets of NHC–Ag complexes. These studies are particularly noteworthy for the discovery of molecular modes of actions, as well as overall pharmacological/toxicological profiles, which in turn facilitate the development of novel metallodrugs.

## Data Availability

Data is contained within the article or Appendix A.

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
