# Peer review of "Antimicrobial Activity and Mode of Action of N-Heterocyclic Carbene Silver(I) Complexes"

_molecules, 2024, doi:10.3390/molecules30010076_

Round 1

Reviewer 1 Report

Comments and Suggestions for Authors

The paper by Zanfardino et al concerns new Ag-NHC (N-heterocyclic carbene) complexes tested for antibiotic agents against Gram +/- bacteria.  This is an active and important area of research, especially for possible use against antibiotic resistant disease.  The authors tested four new complexes, neutral mono-ligated and cationic bi-ligated complexes using two new NHC. The scope of the submission and audience of this work clearly falls into Molecule’s venue, but it has lower novelty, significance and scientific soundness than comparable articles in this journal.
The syntheses and chemical characterizations were straightforward and well presented. The biological assays used were basic and uninformative, given the emphasis on mechanistic interpretation and identification of establishing the targets of Ag-toxicity in the text.  
Dose-response toxicity was determined by simple cell-counting  and MIC at distinct dosages and times. The neutral mono-ligated forms showed higher activity, which was suggested to be due to higher cell uptake due to lipophilicity and lack of charge (as was defined in a previous review by authors). One mono-ligated complex was shown to be much more active, and it was used for further testing. No control experiments using the NHC ligands in the absence of Ag was done.
DAPI/PI staining was used to assess if the treated cells showed cell wall disruption.  None did. SEM was also used to image the treated cells, and likewise showed no gross changes to cells, only diminished numbers after treatment. Neither of these results prove that lipophilicity and lack of charge allowed drug uptake, as uptake was never assessed. More advanced EM can be done to measure the intracellular concentrations and locations of Ag ions, which would allow quantification of uptake and give direct information on possible targets of Ag-toxicity.
Cell survival MTT assays on human HaCaT cells showed toxicity that mimicked that of the bacterial cells, though at higher concentrations. This result is not necessarily good, as any toxicity to human cells is bad for an antibiotic drug candidate! A better experiment would be to use the MTT on the bacteria used in other experiments; although this is less common, optimized methods for use with bacteria has been reviewed (Acta Histochem  2018 May; 120(4):303-311. doi: 10.1016/j.acthis.2018.03.007). Again, no controls of the free NHCs were done.
SEC ICP-MS was used to qualitatively characterize Ag-bound proteins in cytosolic lysates treated cells. Additional experiments were done by adding Ag to untreated cytosolic cell lysates, which showed less Ag binding. Unfortunately, no additional separations or characterizations of these proteins were performed (e.g. by MS, protein sequencing, or pull-down separations).  No discussion of possible protein targets identified in previous studies was given, only the promise of further research on this; then why is this incomplete result included?  
I suggest reconsideration after major revisions. Ag uptake is suggested to be the main determinant of toxicity, but that was not assessed cleanly in any experiment. Further studies, collaborations with cell biologists, and perhaps better design of experiments will be needed to get usable information from this system.

Comments on the Quality of English Language

Overall the English was good, but many odd word choices and misuses throughout. These should be easily corrected with editorial assistance.

Examples:

17 "Currently, due to antibiotic resistance, a new era for silver complexes may open"

18 "In this scenario, N-heterocyclic carbene (NHC) ligands have been applied"

26 "have been reported to gain more insight into the mechanism"

43 "able to tune the reactivity and address Ag(I) toward the target"

45 "The wide diffusion and applications of this moiety is due to easy synthetic"

50  "holding the efficacy on the wound-site."

58 "extracellular media can destroy bacterial membranes"

65/66 "If many studies are reviewed about the mechanism of actions of silver nanoparticles [13-14], few studies are reported about the mechanism of silver complexes"

69 "in Escherichia . coli"

189 "Microscopy essays"

243, 248, 257 "SEC-ICP MS" should be SEC ICP-MS

267 "Chemistry synthesis" should be syntheses

Author Response

Thanks to Reviewer 1 for the important comments.

The paper by Zanfardino et al concerns new Ag-NHC (N-heterocyclic carbene) complexes tested for antibiotic agents against Gram +/- bacteria.  This is an active and important area of research, especially for possible use against antibiotic resistant disease.  The authors tested four new complexes, neutral mono-ligated and cationic bi-ligated complexes using two new NHC. The scope of the submission and audience of this work clearly falls into Molecule’s venue, but it has lower novelty, significance and scientific soundness than comparable articles in this journal.
The syntheses and chemical characterizations were straightforward and well presented. The biological assays used were basic and uninformative, given the emphasis on mechanistic interpretation and identification of establishing the targets of Ag-toxicity in the text.  
Dose-response toxicity was determined by simple cell-counting and MIC at distinct dosages and times. The neutral mono-ligated forms showed higher activity, which was suggested to be due to higher cell uptake due to lipophilicity and lack of charge (as was defined in a previous review by authors). One mono-ligated complex was shown to be much more active, and it was used for further testing. No control experiments using the NHC ligands in the absence of Ag was done.

Thanks to reviewer 1 for comments, we have now included controls using the NHC ligands in the absence of Ag, in all experiments. Furthermore, we tried to expand the biological part.

DAPI/PI staining was used to assess if the treated cells showed cell wall disruption.  None did. SEM was also used to image the treated cells, and likewise showed no gross changes to cells, only diminished numbers after treatment. Neither of these results prove that lipophilicity and lack of charge allowed drug uptake, as uptake was never assessed. More advanced EM can be done to measure the intracellular concentrations and locations of Ag ions, which would allow quantification of uptake and give direct information on possible targets of Ag-toxicity. Cell survival MTT assays on human HaCaT cells showed toxicity that mimicked that of the bacterial cells, though at higher concentrations. This result is not necessarily good, as any toxicity to human cells is bad for an antibiotic drug candidate! A better experiment would be to use the MTT on the bacteria used in other experiments; although this is less common, optimized methods for use with bacteria has been reviewed (Acta Histochem  2018 May; 120(4):303-311. doi: 10.1016/j.acthis.2018.03.007). Again, no controls of the free NHCs were done.

To confirm the passage of the Ag5MC compound through the membrane without damaging it, we performed an assay using a 1-N-phenyl naphthylamine probe. As shown in figure 7, and as explained in the text, it is clear that the bacterial membrane is not compromised and that the targets of our compound are probably cytosolic proteins to which the Ag is able to bind. Furthermore, to confirm the antimicrobial activity of our compounds, we performed, as suggested by the reviewer, the MTT assay directly on the bacteria used in other experiments, see figure 4, (optimizing methods for use with bacteria, Acta Histochem 2018 May; 120(4):303-311. doi: 10.1016/j.acthis.2018.03.007). Also in these experiments we included the controls (controls of the free NHCs compounds).

SEC ICP-MS was used to qualitatively characterize Ag-bound proteins in cytosolic lysates treated cells. Additional experiments were done by adding Ag to untreated cytosolic cell lysates, which showed less Ag binding. Unfortunately, no additional separations or characterizations of these proteins were performed (e.g. by MS, protein sequencing, or pull-down separations).  No discussion of possible protein targets identified in previous studies was given, only the promise of further research on this; then why is this incomplete result included?

In this paper we reported only a preliminary results about the target. In this step of the research we wanted to confirm only the presence of silver ion in the cytosol and his uptake to proteins. In the future, we will join with other expertise for further studies in this field.

I suggest reconsideration after major revisions. Ag uptake is suggested to be the main determinant of toxicity, but that was not assessed cleanly in any experiment. Further studies, collaborations with cell biologists, and perhaps better design of experiments will be needed to get usable information from this system.

Overall the English was good, but many odd word choices and misuses throughout. These should be easily corrected with editorial assistance.

Examples:

17 "Currently, due to antibiotic resistance, a new era for silver complexes may open"

18 "In this scenario, N-heterocyclic carbene (NHC) ligands have been applied"

26 "have been reported to gain more insight into the mechanism"

43 "able to tune the reactivity and address Ag(I) toward the target"

45 "The wide diffusion and applications of this moiety is due to easy synthetic"

50  "holding the efficacy on the wound-site."

58 "extracellular media can destroy bacterial membranes"

65/66 "If many studies are reviewed about the mechanism of actions of silver nanoparticles [13-14], few studies are reported about the mechanism of silver complexes"

69 "in Escherichia . coli"

189 "Microscopy essays"

243, 248, 257 "SEC-ICP MS" should be SEC ICP-MS

267 "Chemistry synthesis" should be syntheses

We followed the reviewers suggestions

Reviewer 2 Report

Comments and Suggestions for Authors

Manuscript Number : molecules-3268972

Title: Antimicrobial activity and mode of action of N-Heterocyclic 2 carbene Silver (I) complexes
In the reviewed manuscript, the authors report on the antimicrobial activity of four N-heterocyclic carbene silver complexes.

The biological activity of the complexes has been evaluated  by: antimicrobial activity (E.coli, S. aureus, C. albicans), microscopy essays, MTT assay, SEC ICP MS analysis. The overall quality of the work is good, however it requires a major revision before it will be accepted for publication. Below is a list of the most important corrections that the authors should make:

1.In section 3.1. the authors present the synthesis of two  ligands and two complexes.

I recommend adding the abbreviation for the ligands and complexes after the real name:

a)       Ligands:

3.1.1 2-(2-acetamidoethyl)-2H-imidazo[1,5-a]pyridin-4-ylium chloride (abbreviation MC or BC ???)

3.1.3Synthesis of 2-pyridin- N-(2-ethylacetylamido)-2-yl-2-imidazole Hexafluorophosphate (abbreviation MC or BC ???)

b)      Complexes:

3.1.2 Synthesis of 2-(2-acetamidoethyl)-2H-imidazo[1,5-a]pyridin-4-ylium silver (I) chloride (Ag5MC ?)

3.1.4 Synthesis of 2-pyridin- N-(2-ethylacetylamido)-2-yl-2-imidazole silver (I)Hexafluorophosphate (Ag5BC ?)

What about the synthesis of the next two complexes ? In section 2.1 the author mentioned that the Ag4MC complex was demonstrated by Liu et al. [Ref. 20], however, in section 3.1 the authors should present the short description of the synthesis and comment that the spectral parameters as 1H NMR stay in good or not agreement with those reported by Liu. Apart from the complex Ag4MC, are three complexes novel ? The structures of the complexes were not determined by using single crystal X-ray analysis. I understand that the authors have a some problems with the crystallization of the complexes. Therefore, I recommend to provide the other evidence such as 13C NMR and IR spectra for ligands and complexes. Also, the figures of the 1H, 13C and IR spectra should be presented as a Supplementary Materials. I am somewhat dissatisfied with the evidence provided regarding for the structure of coordination compounds. This especially applies to the structure in solutions (because biological tests are performed on compounds dissolved in solvents).

2. In the abstract: I recommend adding the abbreviations of the complexes.

3. The Figure 1 – should be moved to the 2.1 Section.

4. In section 2.1 – the author mentioned about Au4MC complex (line 119) but there is no reference.

5. In section 2.1 – ‘’The ligand was synthesized adapting a previously reported procedure [21] (line 124).’’ – My question is: which ligand ? It is unclear.

6. Generally, in my opinion, the section 2.1 should start with the information about the ligands and then the complexes. In the current version, it looks like a mix with any logical order.

7. I strongly recommend adding an abbreviation after the full name of the ligands and complexes wherever possible.

8. In Scheme 1 – is it possible to add the Ag4MC and Ag4BC ? If yes/so, then it is not necessary to present the Figure 1.

9. In order to enrich and extend the manuscript, I suggest to perform molecular docking with microorganisms. For example for the macromolecular targets as receptors: Staphylococcus aureus metalloproteinase from Staphylococcus aureus and a penicillin−binding protein 3 from Escherichia coli to determine test the nature of the Ag(I) complexes−receptor interactions and to identify potential binding modes and energies energy.

Overall, the Ms could be accepted subject to these "major changes".

Comments on the Quality of English Language

The English could be improved to more clearly express the research.

Author Response

Thanks to Reviewer 2 for the important comments.

The biological activity of the complexes has been evaluated by: antimicrobial activity (E.coli, S. aureus, C. albicans), microscopy essays, MTT assay, SEC ICP MS analysis. The overall quality of the work is good, however it requires a major revision before it will be accepted for publication. Below is a list of the most important corrections that the authors should make:
1.In section 3.1. the authors present the synthesis of two ligands and two complexes.

I recommend adding the abbreviation for the ligands and complexes after the real name:

  1. a) Ligands:
    1.1 2-(2-acetamidoethyl)-2H-imidazo[1,5-a]pyridin-4-ylium chloride (abbreviation MC or BC ???) 3.1.3Synthesis of 2-pyridin- N-(2-ethylacetylamido)-2-yl-2-imidazole Hexafluorophosphate (abbreviation MC or BC ???)
  2. b) Complexes:
    1.2 Synthesis of 2-(2-acetamidoethyl)-2H-imidazo[1,5-a]pyridin-4-ylium silver (I) chloride (Ag5MC ?) 3.1.4 Synthesis of 2-pyridin- N-(2-ethylacetylamido)-2-yl-2-imidazole silver (I)Hexafluorophosphate (Ag5BC ?)
    What about the synthesis of the next two complexes ? In section 2.1 the author mentioned that the Ag4MC complex was demonstrated by Liu et al. [Ref. 20], however, in section 3.1 the authors should present the short description of the synthesis and comment that the spectral parameters as 1H NMR stay in good or not agreement with those reported by Liu. Apart from the complex Ag4MC, are three complexes novel ? The structures of the complexes were not determined by using single crystal X-ray analysis. I understand that the authors have a some problems with the crystallization of the complexes. Therefore, I recommend to provide the other evidence such as 13C NMR and IR spectra for ligands and complexes. Also, the figures of the 1H, 13C and IR spectra should be presented as a Supplementary Materials. I am somewhat dissatisfied with the evidence provided regarding for the structure of coordination compounds. This especially applies to the structure in solutions (because biological tests are performed on compounds dissolved in solvents).

We followed the reviewer’s suggestions. We carried out elemental analysis, IR and 13C NMR spectra. Furthermore, we have added all spectra of the new compounds in the supplementary materials. We could not get the crystal suitable for X-rays.

  1. In the abstract: I recommend adding the abbreviations of the complexes.

We thank the reviewer and we added the abbreviations

  1. The Figure 1 – should be moved to the 2.1 Section.

We moved the figure in the 2.1 section

  1. In section 2.1 – the author mentioned about Au4MC complex (line 119) but there is no reference.

The reference was recalled at line 119

  1. In section 2.1 – ‘’The ligand was synthesized adapting a previously reported procedure [21] (line 124).’’ – My question is: which ligand ? It is unclear.

We thank the reviewer for the suggestion to better specify which ligand we are talking about

  1. Generally, in my opinion, the section 2.1 should start with the information about the ligands and then the complexes. In the current version, it looks like a mix with any logical order.

We thank the reviewer for thois suggestion and now we first described the 4MC and 4BC ligands and the Ag4MC and Ag4BC complexes chosen from the literature.  Therefore we reported the 5MC and 5BC ligands and the Ag5MC and Ag5BC complexes

  1. I strongly recommend adding an abbreviation after the full name of the ligands and complexes wherever possible.

We added the abbreviation wherever possible.

  1. In Scheme 1 – is it possible to add the Ag4MC and Ag4BC ? If yes/so, then it is not necessary to present the Figure

 We agree to the reviewer to simplify the figures but in the scheme 1 only the synthesis reaction of the Ag5MC and Ag5BC complexes is shown. Therefore we believe should be better to separate Ag4MC and Ag4BC complexes reporting their structure in another figure. 

  1. In order to enrich and extend the manuscript, I suggest to perform molecular docking with microorganisms. For example for the macromolecular targets as receptors: Staphylococcus aureus metalloproteinase from Staphylococcus aureus and a penicillin−binding protein 3 from Escherichia coli to determine test the nature of the Ag(I) complexes−receptor interactions and to identify potential binding modes and energies energy.

Overall, the Ms could be accepted subject to these "major changes".

Thanks to reviewer 2 for comment, we intend to further study the mechanism of action of the compound Ag5MC, thus identifying the proteins to which the Ag binds, preventing them from carrying out their biological functions and leading to the death of the microorganisms. In our future experiments we will certainly include some molecular docking experiments.

Reviewer 3 Report

Comments and Suggestions for Authors

Manuscript presents the preparation of four Ag(I) NHC complexes and extended bioevaluation. Chemical part is rather weak. Authors report only 1H NMR and MS-molecular peak. This is certainly not enough. Elemental analysis should be reported, IR and 13C NMR as well and results of HRMS with enough digits. It would be very valuable if at least some proposed structures would be supported by SC XRD as a superior technique in inorganic/organometallic chemistry. Biological part is of interest and seems to be well performed – antimicrobial activity, microscopy assays, MTT assay and ICP MS analysis of cytosolic proteins was performed. These results are of interest.

Authors should extend the chemical component of this paper as written above, than this contribution could become suitable for publication.

Other minor issues:

-          authors should take care when to use capital letters. In a title “Heterocyclic” and “Silver” should not have capital letters. Also, in the text “N-Heterocyclic Carbenes”, “N-Acyl Homoserine Lactone Lactonase” should not have capital letters.

-          IUPAC nomenclature should be followed – oxidation state should be given without a space between a metal and oxidation state – for example “silver(I)” and not “silver (I)”. There are many such typo errors throughout the text.

-          “Escherichia . coli” should be without the dot or change into E. coli. “E. coli.” should be “E. coli”

-          letter H in names such as “2H-imidazole” should be italic.

-          “chloro…silver(I)” should be “chlorido…silver(I)” and similar for bromo/bromido.

-          Scheme 1: reorient the first structure in order to be the amide group on the left side as in all other structures – in that way it will be more reader friendly. Also, in four structures redraw the amide group in order not to have 180° angle at –NH– moiety. Also, unify the font – use only Arial (at least for all elements).

-          literature list is rather modest with only 2 references – there are certainly many more important works that should be mentioned here in order to give suitable overview on this interesting and promising area of research.

-          Reference style should be unified: some journal names are abbreviated and some are not. In some abbreviated journal names dots are used while in some are not. Also, in ref. 26 journal name is missing (The Royal Society of Chemistry is not a journal name, it should be Chemical Science)

Author Response

Thanks to the Reviewer 3 for the important comments.

Manuscript presents the preparation of four Ag(I) NHC complexes and extended bioevaluation. Chemical part is rather weak. Authors report only 1H NMR and MS-molecular peak. This is certainly not enough. Elemental analysis should be reported, IR and 13C NMR as well and results of HRMS with enough digits. It would be very valuable if at least some proposed structures would be supported by SC XRD as a superior technique in inorganic/organometallic chemistry. Biological part is of interest and seems to be well performed – antimicrobial activity, microscopy assays, MTT assay and ICP MS analysis of cytosolic proteins was performed. These results are of interest.

We followed the reviewer’s suggestions. We carried out elemental analysis, IR, 13C NMR spectra, 19F NMR and 31P NMR. Furthermore, we have added all spectra of the new compounds in the supplementary materials. We could not get crystals suitable for X-rays.

Authors should extend the chemical component of this paper as written above, than this contribution could become suitable for publication.

Other minor issues:

-          authors should take care when to use capital letters. In a title “Heterocyclic” and “Silver” should not have capital letters. Also, in the text “N-Heterocyclic Carbenes”, “N-Acyl Homoserine Lactone Lactonase” should not have capital letters.

We followed the reviewers suggestions

-          IUPAC nomenclature should be followed – oxidation state should be given without a space between a metal and oxidation state – for example “silver(I)” and not “silver (I)”. There are many such typo errors throughout the text.

-          “Escherichia . coli” should be without the dot or change into E. coli. “E. coli.” should be “E. coli”

We followed the reviewers suggestions

-          letter H in names such as “2H-imidazole” should be italic.

We followed the reviewers suggestion

-          “chloro…silver(I)” should be “chlorido…silver(I)” and similar for bromo/bromido.

Both chloro…silver(I)”  and “chlorido…silver(I) are accepted by IUPAC. In the previous paper in literature the complex Ag4MC is reported as chloro…silver(I), therefore we chose this form.

-          Scheme 1: reorient the first structure in order to be the amide group on the left side as in all other structures – in that way it will be more reader friendly. Also, in four structures redraw the amide group in order not to have 180° angle at –NH– moiety. Also, unify the font – use only Arial (at least for all elements).

We modified the scheme 1

-          literature list is rather modest with only 2 references – there are certainly many more important works that should be mentioned here in order to give suitable overview on this interesting and promising area of research.

-          Reference style should be unified: some journal names are abbreviated and some are not. In some abbreviated journal names dots are used while in some are not. Also, in ref. 26 journal name is missing (The Royal Society of Chemistry is not a journal name, it should be Chemical Science).

We followed the reviewers suggestions

Round 2

Reviewer 1 Report

Comments and Suggestions for Authors

This resubmission is considerably improved from the original. My only request is that the authors include some evaluation or conclusion regarding their testing of the toxicity of Ag5MC to human HaCaT cells in Fig. 8 in comparison to the toxicity against bacteria cell lines shown in Fig. 3. This evaluation could be in section 2.4 describing the experiments, or in the Conclusion section 4. My interpretation is that the LD50 concentration is roughly 20 fold lower against S. aureus and C. albicans, than HaCaT, and more than 200 fold lower against E. coli.

Author Response

Reviewer 1 Round 2

Manuscript Number : molecules-3268972

Title: Antimicrobial activity and mode of action of N-Heterocyclic 2 carbene Silver (I) complexes

This resubmission is considerably improved from the original. My only request is that the authors include some evaluation or conclusion regarding their testing of the toxicity of Ag5MC to human HaCaT cells in Fig. 8 in comparison to the toxicity against bacteria cell lines shown in Fig. 3. This evaluation could be in section 2.4 describing the experiments, or in the Conclusion section 4. My interpretation is that the LD50 concentration is roughly 20 fold lower against S. aureus and C. albicans, than HaCaT, and more than 200 fold lower against E. coli.

Thanks to reviewer 1 for the comment, it has been included in paragraph 2.4.

Reviewer 2 Report

Comments and Suggestions for Authors

as an attachment

Comments on the Quality of English Language

The English could be improved to more clearly express the research.

Author Response

Reviewer 2 Round 2

Manuscript Number : molecules-3268972

Title: Antimicrobial activity and mode of action of N-Heterocyclic 2 carbene Silver (I) complexes

In the reviewed manuscript, the authors report on the antimicrobial activity of four N-heterocyclic carbene

silver complexes.

The authors responded mostly positively to the criticism and have revised of their manuscript. I still have

some suggestions / doubts.

General remark: it is possible to designate ligands (abbreviation) as simple L1, L2 etc and complexes as 1, 2,

3 and 4. The formulae of the complexes are also necessary, for example Ag5MC => [Ag(L)Cl] where L = 2-(2-

acetamidoethyl)-2H-imidazo[1,5-a]pyridin-4-ylium).

We reported in text only twice the IUPAC name of the ligand for this reason we did not need the abbreviation

  1. Line 111 and 112 For these statements, we designated the bromo[1,3-diethyl-4,5-bis(4-

methoxyphenyl)imidazol-2- 112 ylidene]silver(I) (Ag4MC) complex [20]. y opinion, this is not a desinge

but a synthesis according the procedure described in ref [20].

We took up the suggestion of the reviewer by replacing design with selected

2.Line 118: The cationic 116 (bis[1,3-diethyl-4,5-bis(4-methoxyphenyl)imidazol-2-ylidene]silver(I)(Ag4BC)

complex was prepared to compare the activity of the positive charge of this compound to Au4MC.

What is the compund Au4MC ? Is it Ag4MC ? - and the comparison of Ag4BC and Ag4MC makes sense in

this manuscript.

We apologize AuMC is a mistyping the complex is Ag4MC

  1. Among the four complexes Ag4MC + Ag4BC and Ag5MC + Ag5BC two of them (Ag5MC + Ag5BC) are new.

The complex Ag4MC was synthesized according the procedudure from ref [20]. For the complex Ag4BC

there is no reference and the procedure of synthesis is not given anywehere in the manuscript.

Line 118 The cationic 116 (bis[1,3-diethyl-4,5-bis(4-methoxyphenyl)imidazol-2-ylidene]silver(I)(Ag4BC) [

reference ??]

The reference is the same of Ag4MC [20]. Now we added the reference also for Ag4BC

  1. In Scheme 1 is it possible to add the Ag4MC and Ag4BC ? If yes/so, then it is not necessary to present the

Figure

Authors response: We agree to the reviewer to simplify the figures but in the scheme 1 only the synthesis

reaction of the Ag5MC and Ag5BC complexes is shown. Therefore we believe should be better to separate

Ag4MC and Ag4BC complexes reporting their structure in another figure.

There is no other figure with Ag4MC and Ag4BC. The authors have not presented this aspect in another

figure in main manuscript or in the Sup. Info.

The chemical structures of Ag4MC and Ag4BC are reported in figure 1

  1. Sup. Info. The figures 10 and 14 present the same spectrum, 13C NMR of Ag5BC.

The figures 10 and 14 show two different spectra. The chemical shift are quite similar due their structure.

  1. Line 319 The spectra were obtained by using a Ge single crystal at a resolution of 4 cm 1 in attenuated

total reflection mode on a Jasco FT/IR 4100 spectromete... Does this mean FT-IR spectra ?

That’s right, now we have inserted FT-IR before spectra

  1. Line 323-325 The complexes bromo[1,3-diethyl-4,5-bis(4-methoxy- phenyl)imidazol-2-ylidene]silver(I)

(Ag4MC) and bis[1,3-diethyl-4,5-bis(4-methoxy- phenyl)imidazol-2-ylidene]silver(I) (Ag4BC) were

synthesized as reported in literature [20]. Should be moved to the 3.1 Syntheses part or somewhere in

2.1. Selection and synthesis of the complexes part => for clarity.

We duplicated the sentence in section 2.1

  1. Sup. Info => The authors have not provided any evidences for the complexes Ag4MC and Ag4BC (1H, 13C,

IR, mas spectra and etc.). As a readers we are not sure that the authors synthesized the same complexes as

in ref. [20].

We carried out mass spectra and 1H NMR of both complexes to confirm the identity of the complexes as reported in literature. The mass spectra showed only the mass of the ligand due to fragmentation as reported in literature [20].

Reviewer 3 Report

Comments and Suggestions for Authors

Manuscript is now much improved and is suitable for publication. There are atill the same typo errors: in Title, on pages 2, 3 and 11 "silver (I)" should be corrected to "silver(I)"

Author Response

Reviewer 3 Round 2

Manuscript Number: molecules-3268972

Title: Antimicrobial activity and mode of action of N-Heterocyclic 2 carbene Silver (I) complexes

Manuscript is now much improved and is suitable for publication. There are atill the same typo errors: in Title, on pages 2, 3 and 11 "silver (I)" should be corrected to "silver(I)"

Thanks to reviewer 3, the typo errors have been corrected.